# Trends in the Prescription of Non-Vitamin K Antagonist Oral Anticoagulants for Atrial Fibrillation: Results of the Polish Atrial Fibrillation (POL-AF) Registry

**DOI:** 10.3390/jcm9113565

**Published:** 2020-11-05

**Authors:** Iwona Gorczyca, Olga Jelonek, Beata Uziębło-Życzkowska, Magdalena Chrapek, Małgorzata Maciorowska, Maciej Wójcik, Robert Błaszczyk, Agnieszka Kapłon-Cieślicka, Monika Gawałko, Monika Budnik, Tomasz Tokarek, Renata Rajtar-Salwa, Jacek Bil, Michał Wojewódzki, Anna Szpotowicz, Janusz Bednarski, Elwira Bakuła-Ostalska, Anna Tomaszuk-Kazberuk, Anna Szyszkowska, Marcin Wełnicki, Artur Mamcarz, Beata Wożakowska-Kapłon

**Affiliations:** 1Collegium Medicum, The Jan Kochanowski University, 25-369 Kielce, Poland; iwona.gorczyca@interia.pl (I.G.); olga_jelonek@wp.pl (O.J.); bw.kaplon@poczta.onet.pl (B.W.-K.); 21st Clinic of Cardiology and Electrotherapy, Swietokrzyskie Cardiology Centre, 25-736 Kielce, Poland; 3Department of Cardiology and Internal Diseases, Military Institute of Medicine, 04-141 Warsaw, Poland; mmaciorowska@wim.mil.pl; 4Faculty of Natural Sciences, The Jan Kochanowski University, 25-369 Kielce, Poland; chrapek.magdalena@gmail.com; 5Department of Cardiology, Medical University of Lublin, 20-059 Lublin, Poland; m.wojcik@umlub.pl (M.W.); robertblaszczyk1@wp.pl (R.B.); 61st Chair and Department of Cardiology, Medical University of Warsaw, 02-097 Warsaw, Poland; agnieszka.kaplon@gmail.com (A.K.-C.); mongawalko@gmail.com (M.G.); moni.budnik@gmail.com (M.B.); 7Department of Cardiology and Cardiovascular Interventions, University Hospital, 30-688 Krakow, Poland; tomek.tokarek@gmail.com (T.T.); rajfura@op.pl (R.R.-S.); 8Department of Invasive Cardiology, Centre of Postgraduate Medical Education, Central Clinical Hospital of the Ministry of Interior and Administration, 02-507 Warsaw, Poland; biljacek@gmail.com (J.B.); michaljerzywojewodzki@gmail.com (M.W.); 9Department of Cardiology, Regional Hospital, 27-400 Ostrowiec Swiętokrzyski, Poland; szpotowiczanna@wp.pl; 10Department of Cardiology, St John Paul II Western Hospital, 05-825 Grodzisk Mazowiecki, Poland; medbed@wp.pl (J.B.); elwira.bakula@gmail.com (E.B.-O.); 11Department of Cardiology, Medical University, 15-276 Bialystok, Poland; a.tomaszuk@poczta.fm (A.T.-K.); annaszyszkowska92@gmail.com (A.S.); 123rd Department of Internal Diseases and Cardiology, Warsaw Medical University, 02-091 Warsaw, Poland; welnicki.marcin@gmail.com (M.W.); artur.mamcarz@wum.edu.pl (A.M.)

**Keywords:** atrial fibrillation, non-vitamin K antagonist oral anticoagulants, oral anticoagulants, vitamin K antagonists

## Abstract

Background: Current guidelines do not suggest in which groups of patients with atrial fibrillation (AF) individual non-vitamin K antagonist oral anticoagulants (NOACs) should be used for the prevention of thromboembolic complications. The aim of this study was to evaluate the frequency of use of apixaban, dabigatran, and rivaroxaban, and attempt to identify factors predisposing their administration. Methods: The Polish Atrial Fibrillation (POL-AF) registry is a prospective, non-interventional study, including consecutive patients with AF hospitalized in ten Polish cardiology centers during the period ranging from January to December 2019. In this study, all patients were treated with NOACs. Results: Among the 2971 patients included in the analysis, 40.4% were treated with rivaroxaban, 32% with apixaban, and 27.6% with dabigatran. The mean age of the total population was 72 ± 11.5 years and 43% were female. A reduced dose of NOAC was used in 35% of patients treated with apixaban, 39.7% of patients treated with dabigatran, and 34.4% of patients treated with rivaroxaban. Independent predictors of the use of apixaban were previous bleeding (OR 2.37, CI 1.67–3.38), GFR < 60 mL/min (OR 1.38, CI 1.25–1.64), heart failure (OR 1.38, CI 1.14–1.67) and age (per 5 years) (OR 1.14, CI 1.09–1.19). GFR < 60 mL/min (OR 0.79, CI 0.66–0.95), female (OR 0.8, CI 0.67–0.96) and age (per 5 years) (OR 0.95, CI 0.91–0.99) diminished the chance of using dabigatran. Previous bleeding (OR 0.43, CI 0.28–0.64), vascular disease (OR 0.84, CI 0.70–0.99), and age (per 5 years) (OR 0.94, CI 0.90–0.97) diminished the chance of choosing rivaroxaban. Conclusions: In hospitalized patients with AF, the most frequently chosen NOAC was rivaroxaban. Apixaban was chosen more often in patients after bleeding, and in those who were advanced in years, with heart failure and impaired renal function. Impaired renal function and female gender were factors that diminished the chance of using dabigatran. Previous bleeding and vascular disease was the factor that diminished the chance of using rivaroxaban. Dabigatran and rivaroxaban have been used less frequently in elderly patients.

## 1. Introduction

The prevention of thromboembolic complications is an important part of the management of patients with atrial fibrillation (AF) [1]. European and American guidelines recommend the use of non-vitamin K antagonist oral anticoagulants (NOACs) over therapy with vitamin K antagonists (VKAs) in most AF patients [2,3]. The number of patients treated with NOACs has increased significantly during the last few years [4,5]. In primary randomized controlled trials leading to their approval, compared to warfarin, NOACs were shown to be either non-inferior or superior for stroke prevention in AF, with similar or reduced rates of bleeding, especially intracranial hemorrhage [6,7,8]. However, no head-to-head comparison between the individual NOACs has been performed. Additionally, there were differences in the study populations of each of the pivotal NOAC trials. In the absence of randomized clinical trials, observational studies utilizing the data from clinical practice may add useful information regarding individual NOACs. The proper use of specific NOACs for stroke prevention in AF patients requires a diligent approach in various settings of daily clinical practice. In the currently binding guidelines referring to the treatment of patients with AF, there is a lack of recommendations concerning the choice of individual NOACs in certain cohorts of patients [4,5]. However, in expert documents, there are guidelines regarding the choice of a specific NOAC in various clinical situations [9,10,11]. With the increased availability of NOACs, the prescription patterns and factors driving treatment may evolve.

The aim of this study was to assess the use frequency of apixaban, dabigatran, and rivaroxaban and the predictors of their prescription in a nationwide cohort of hospitalized patients with AF.

## 2. Materials and Methods

### 2.1. Study Design and Study Population

The Polish Atrial Fibrillation (POL-AF) registry is a prospective, observational, multicenter study, whereby consecutive AF patients are enrolled across ten cardiology hospital centers, which represents the Polish cardiology reality well, covering seven academic centers, two district hospitals, and one military hospital. The study was registered in ClinicalTrials.gov: NCT04419012. The aim of the registry was to gain detailed insights into the clinical characteristics and management of patients with AF, especially into the prevention of thromboembolic events. The data were collected from January to December 2019, for two whole weeks out of each month. The registry includes all consecutive patients with AF hospitalized in a participating center during the study period who were hospitalized for urgent and planned reasons. Patients were included if they were at least 18 years of age and had a history of AF documented by electrocardiography or in their medical history. No explicit exclusion criteria were defined to avoid biased selection of patients and to achieve a cohort close to “real life”. Furthermore, consecutive patients were included in each site in order to reduce selection bias. Only patients admitted to hospital to have AF ablation were excluded from the registry because not all of the centers perform catheter ablation. Further, patients undergoing ablation due to AF have a clinical profile that differs from most patients with AF (they are younger and do not have concomitant diseases).

In the presented study, based on the results of the POL-AF registry, patients with AF treated with NOACs were evaluated. Patients receiving VKAs, antiplatelet therapy, and those not being given anticoagulant therapy were excluded from the study. During the study period, 3999 patients with AF were included in the POL-AF registry. After applying the exclusion criteria described above, a total of 2971 patients were included in this study (Figure 1).

### 2.2. Covariates

Investigators collected baseline characteristics regarding demographics, medical history, type of AF, diagnostic test results, and pharmacotherapy.

Thromboembolic risk was defined according to a combined congestive heart failure, hypertension, age ≥75 years, diabetes mellitus, stroke/transient ischemic attack, vascular disease, age 65–74 years, sex category (CHA_2_DS_2_-VASc) score [12]. Bleeding risk was assessed according to a hypertension, abnormal renal/liver function, stroke, bleeding, labile INR (international normalized ratio), elderly (>65 years), drug/alcohol consumption (HAS-BLED) score [13].

The glomerular filtration rate (GFR), which is used to assess patients’ kidney function, was calculated using the CKD-EPI (Chronic Kidney Disease Epidemiology Collaboration) equation.

The study was approved by the Ethics Committee of the Swietokrzyska Medical Chamber in Kielce (104/2018). The Ethics Committee waived the requirement of obtaining informed consent from the patients.

### 2.3. Statistical Analyses

Continuous data were described by means and standard deviations, whereas categorical data were summarized by frequencies and percentages. Group comparisons were performed using the chi-squared or Fisher’s exact test for categorical variables, while one-way ANOVA was used for continuous variables. Odds ratios (OR) with 95% confidence intervals (95% CI) were calculated in logistic regression models. Participating centers were included in multivariable logistic regression models as potential confounders. Two-tailed *p*-values <0.05 were considered statistically significant. All statistical analyses were performed using the R software package version 3.6.2 (R Foundation for Statistical Computing, Vienna, Austria).

## 3. Results

### 3.1. Patient Characteristics

Among the 2971 patients included in the analysis, 1199 (40.4%) were treated with rivaroxaban, 953 (32%) with apixaban, and 819 (27.6%) with dabigatran. Apixaban was selected for the highest percentage of patients in 3 centers and rivaroxaban in 7 centers (Appendix A). The mean age of the total population was 72 ± 11.5 years and 43% were female. Patients on apixaban were older (74.9 ± 11.5 years) compared with patients on dabigatran (70.3 ± 11.1 years) and rivaroxaban (70.8 ± 11.3 years) (*p* < 0.0001). Patients on apixaban were more likely to be female (47.1%) compared with patients on dabigatran (37.4%) and rivaroxaban (43.5%) (*p* < 0.0001). Non-permanent AF was diagnosed in 76.1% of patients, most often in patients treated with rivaroxaban (79.4%). Baseline characteristics of the study population are summarized in Table 1. Renal dysfunction, defined as GFR < 60 mL/min, was diagnosed in 50.3% of patients, and most often in patients treated with apixaban (61.5%). Table 2 illustrates the laboratory parameters and echocardiographic measurements in patients treated with a specific NOAC. In the study group, patients were most often hospitalized to have electrical cardioversion (27.5%) and due to heart failure (20%).

### 3.2. Thromboembolic Risk, Bleeding Risk, and Antithrombotic Therapy Use

Thromboembolic and bleeding risks according to a specific NOAC treatment are reported in Table 1. Patients treated with apixaban had the highest thromboembolic risk (CHA_2_DS_2_-VASc mean± SD 2.7 ± 1.3) and bleeding risk (HAS-BLED mean± SD 2.2 ± 0.9) as compared with patients treated with dabigatran or rivaroxaban (both *p* < 0.0001). Additionally, they had the highest prevalence of most thromboembolic and bleeding risk factors. Figure 2 shows the prescription patterns for specific NOACs based on their CHA_2_DS_2_-VASc scores.

In the study group, 36% of patients were treated with a reduced NOAC dose. Among 1071 patients who were treated with the reduced dose of NOACs, inappropriately reduced doses were observed in 242 patients (22.6%). The reduced dose of NOACs was used in 35% of apixaban patients, 39.7% of dabigatran patients, and 34.4% of rivaroxaban patients (*p* = 0.037). In most patients (81.8%), the same NOAC as recommended at discharge was used before hospitalization (Table 3).

### 3.3. Predictors of the Use of Individual NOACs

In the analysis of individual NOAC selection, logistic regression models were created for apixaban versus dabigatran and rivaroxaban, dabigatran versus apixaban and rivaroxaban, and rivaroxaban versus apixaban and dabigatran.

In the univariate logistic regression analysis, numerous predictors of a specific NOAC choice were found (Appendix A).

In each of the multivariable models, factors associated with the selection of an individual NOAC versus another NOAC were similar, and included age, heart failure, vascular disease, female, non-permanent AF, malignancy, any previous bleeding, GFR < 60 mL/min, and antiplatelet therapy with NOACs.

Table 4 demonstrates predictors of the use of particular antithrombotic drugs. Independent predictors of the use of apixaban were previous bleeding (OR 2.37, CI 1.67–3.38), GFR < 60 mL/min (OR 1.38, CI 1.25–1.64), heart failure (OR 1.38, CI 1.14–1.67), and age (per 5 years) (OR 1.14, CI 1.09–1.19). GFR < 60 mL/min (OR 0.79, CI 0.66–0.95), female (OR 0.8, CI 0.67–0.96) and age (per 5 years) (OR 0.95, CI 0.91–0.99) diminished the chance of using dabigatran. Previous bleeding (OR 0.43, CI 0.28–0.64), vascular disease (OR 0.84, CI 0.70–0.99), and age (per 5 years) (OR 0.94, CI 0.90–0.97) diminished the chance of choosing rivaroxaban.

## 4. Discussion

The POL-AF registry provides an important view of contemporary antithrombotic therapy in patients with AF. The major findings of the present study are as follows. Firstly, our country-specific registry data showed that the most frequently chosen NOAC was rivaroxaban. Secondly, patients treated with apixaban had the highest CHA_2_DS_2_-VASc and HAS-BLED scores. Thirdly, factors predisposing the choice of a particular NOAC were identified.

NOACs have radically changed the management of AF patients, improving both life expectancy and life quality [14]. The frequency of choosing particular NOACs in the prevention of thromboembolic complications in patients with AF depends on the geographical region, clinical characteristics of patients, and doctors’ preferences. It is also possible to observe a change in anticoagulant therapy trends longitudinally, which is connected with the publication of consecutive studies evaluating efficacy and safety of individual NOACs. Dabigatran was the first NOAC available in Poland, rivaroxaban was the second, and apixaban was the third, all of which were available during the whole study period. Edoxaban has been registered in Europe as a drug for preventing thromboembolic complications in patients with AF, however it is not available in Poland. In the present study including hospitalized patients with AF, rivaroxaban was used in 40% of patients treated with NOACs. It was also the most frequently used NOAC in the retrospective population-based cohort of patients with AF [15]. The data from the NCDR PINNACLE registry showed that rivaroxaban was used more commonly compared with dabigatran and apixaban [16]. Additionally, the data from the Eurobservational Research Program on Atrial Fibrillation (EORP-AF) also indicates it as the most often chosen NOAC [17].

The National Danish Patient Registry cohort included patients from the years 2012 to 2015. This study showed an increase in apixaban use since its introduction and a decline in dabigatran use [18]. The highest proportion of patients treated with apixaban was also declared in The Norwegian Patient Registry [19]. In the present study, a reduced NOAC dose was most often administered to patients treated with dabigatran, just as in the Norwegian Patient Registry [20].

In the present study, in the assessment of factors indicating the choice of a particular NOAC, participating centers were included as potential confounders. The influence of a center on the selection of a specific NOAC may be related to local familiarity with these drugs—some sites may have more exposure to and experience with a specific NOAC. When selecting among NOACs, in the absence of robust head-to-head data, it is possible that local site factors are significant. Physicians tend to select those NOACs with which they are most comfortable and knowledgeable.

As the previous study reported, a high percentage of patients with low thromboembolic risk (CHA_2_DS_2_-VASc 0 in males or 1 in females) were treated with OACs [20,21,22]. In the present study, 93% of patients were at high risk of thromboembolic complications according to their CHA_2_DS_2_-VASc score. By extension, the proportion of low-risk patients receiving NOACs was not high, and most of them were patients with a temporal indication for OACs (before or after ablation or cardioversion).

In the present study, the mean CHA_2_DS_2_-VASc scores were 4.6, 4.2, and 4.2 for patients treated with apixaban, dabigatran, and rivaroxaban, respectively. In the XANTUS (Xarelto for Prevention of Stroke in Patients With Atrial Fibrillation) (registry the mean CHA_2_DS_2_-VASc score was 3.4 for rivaroxaban-treated patients [23], while in the APAF registry the mean CHA_2_DS_2_-VASc score was 3.8 for apixaban-treated patients [24].

In the present study, patients receiving apixaban had the highest CHA_2_DS_2_-VASc and HAS-BLED scores in comparison to patients treated with dabigatran and rivaroxaban; this was probably a reflection of the increased risk of heart failure in these patients and their older age.

The same results were obtained in the study by Maura et al. [25] with a group of 127,841 patients with AF who were administered NOACs. In the Norwegian Patient Registry [19], patients taking apixaban had the highest CHA_2_DS_2_-VASc scores, whereas the highest HAS-BLED scores were reported in patients receiving rivaroxaban.

In the analysis of factors influencing the choice of particular NOACs, the components of the CHA_2_DS_2_-VASc and HAS-BLED scores, and not their results, were taken into account. This is due to the fact that the same results for the aforementioned scores can present in completely different patients. Furthermore, expert documents suggesting the selection of individual NOACs also indicate clinical features, and not results in the scores, as factors relevant to the choice of a particular pharmaceutical.

In the present study, elderly age was a factor predisposing choice of apixaban. In the ARISTOTLE (Apixaban for the Prevention of Stroke in Subjects With Atrial Fibrillation) study, the patients were younger than in the RE-LY (Randomized Evaluation of Long Term Anticoagulant Therapy) and ROCKET-AF (The Rivaroxaban Once Daily Oral Direct Factor Xa Inhibition Compared with Vitamin K Antagonism for Prevention of Stroke and Embolism Trial in Atrial Fibrillation)studies [6,7,8]. However, in the ARISTOTLE study, it was shown that apixaban was equally effective in all age groups [6]. In the subgroup of 88,582 very old ( ≥ 80 y) patients from the ARISTOPHANES (Clinical and Economic Outcomes of Oral Anticoagulants in Non-valvular Atrial Fibrillation) study, apixaban was associated with a lower risk of stroke, systemic embolism, and major bleeding compared with dabigatran and rivaroxaban [26]. Similarly, in a real-world study of 264,479 patients, it was shown that among elderly AF patients, apixaban was associated with significantly lower risks of all-cause, stroke, or systemic embolism-related and major bleeding-related hospitalizations compared with warfarin, dabigatran, and rivaroxaban [27].

As in the POL-AF registry, in the PAROS study of 2027 AF patients, apixaban was more likely than other NOACs to be prescribed in older patients after bleeding and with decreased renal function [28]. In the ORBIT-AF II study, elderly age also predisposed the choice of apixaban vs. rivaroxaban [29].

Previous hemorrhagic complications are a significant factor influencing the withdrawal of OACs in the prevention of thromboembolic complications [30]. In the present study, previous bleeding was a predisposing factor for use of apixaban, which reduced the chance of using rivaroxaban. In a nationwide study of patients with AF in Norway, it was found that dabigatran and apixaban were both associated with a significantly lower risk of major bleeding compared with rivaroxaban [19]. The data from the Danish nationwide registry showed that rivaroxaban was associated with a higher risk of major bleeding compared with apixaban [31].

Decreased renal function is a recognized risk factor for thrombus formation, stroke, systemic embolism, and bleeding events [32,33]. There are also some acknowledged data pointing out that NOACs could reduce the risk of stroke or systemic embolism and major bleeding with respect to different levels of renal function [34,35]. Medicare data showed that apixaban, compared with warfarin, was associated with a decreased risk of stroke or systemic embolism and major bleeding. Risks for both outcomes with rivaroxaban and dabigatran did not differ from risks with warfarin in patients with impaired renal function [36]. In the present study, GFR < 60 mL/min/1.73 m^2^ was a factor predisposing the choice of apixaban and diminishing the chance of using rivaroxaban or dabigatran.

Although current guidelines make no distinction between non-permanent and permanent AF for stroke prevention, there are clinical data confirming that the type of AF is associated with an increased risk of stroke [37]. Therefore, it is possible that in the future, the type of AF will be taken into consideration in the stratification of thromboembolic risk and in choosing anticoagulant prophylaxis. In the presented study, type of AF was not a predisposing factor for apixaban, dabigatran, or rivaroxaban.

## 5. Limitations

Several limitations related to the nature of the data used should be underlined. First of all, due to the lack of long-term observation of patients, it was not possible to evaluate the long-term prognosis of patients with AF treated with individual NOACs. Secondly, in the present study, hospitalized patients with AF were assessed; among these, only some had a first-time diagnosis of AF and only in these patients was an anticoagulant therapy initiated. Thus, despite the registry referring to hospitalized patients, anticoagulant therapy for most of them was initiated in ambulatory conditions before hospital admission. In our study, the data related to edoxaban were not presented because this drug is not available in Poland.

Patients admitted to hospital to have AF ablation were excluded from the registry for two reasons. Firstly, not all centers perform catheter ablation. Secondly, it was acknowledged that patients undergoing ablation due to AF have a clinical profile different from most patients with AF (they are younger and do not have concomitant diseases).

Nevertheless, our data present a comprehensive picture of current Polish AF patients and cardiologist practices, which will provide useful and reliable insights into real-world clinical practice.

## 6. Conclusions

The POL-AF registry shows a full picture of the contemporary use of NOACs in AF patients. In hospitalized patients with AF, the most frequently chosen NOAC was rivaroxaban.

Apixaban was chosen more often in patients after bleeding, and in those who were advanced in years, with heart failure and impaired renal function. Impaired renal function and female gender were factors that diminished the chance of using dabigatran. Previous bleeding and vascular disease was the factor that diminished the chance of using rivaroxaban. Dabigatran and rivaroxaban have been used less frequently in elderly patients. The above results may be of great importance in clinical practice due to the lack of data referring to NOAC application in individual clinical situations.

## Figures and Tables

**Figure 1 jcm-09-03565-f001:**
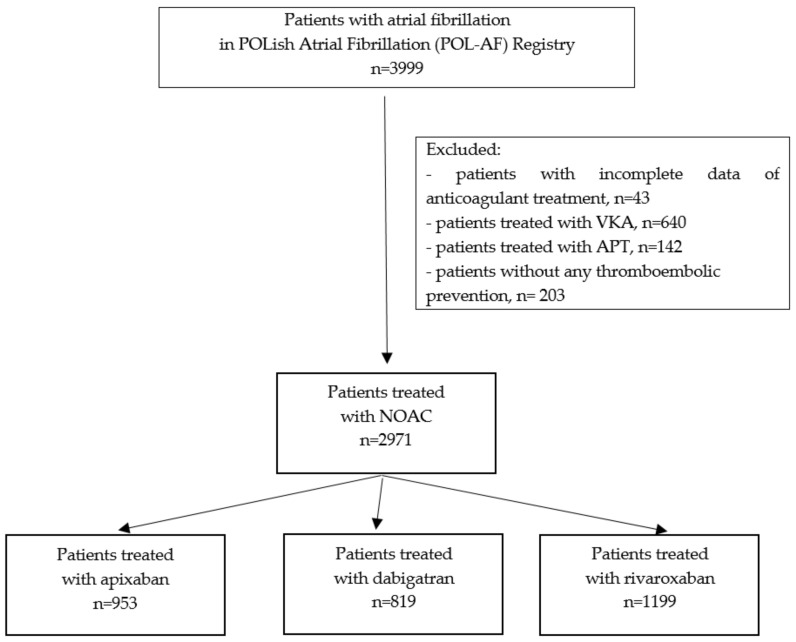
The flow chart of the study. Abbreviations: APT, antiplatelet drug; NOAC, non-vitamin K antagonist oral anticoagulant; VKA, vitamin K antagonist.

**Figure 2 jcm-09-03565-f002:**
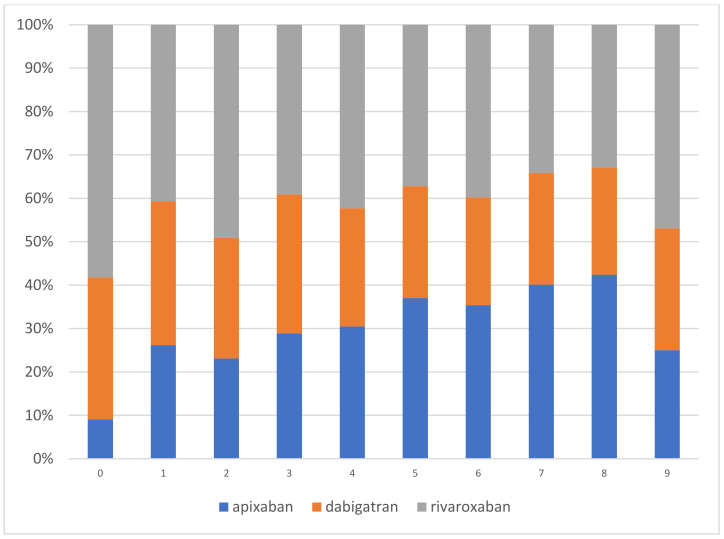
The prescription patterns for specific NOACs based on their CHA2DS2-VASc scores.

**Table 1 jcm-09-03565-t001:** Clinical characteristics of patients treated with apixaban, dabigatran, and rivaroxaban.

Clinical Characteristic	All *n* = 2971	Apixaban *n* = 953	Dabigatran *n* = 819	Rivaroxaban *n* = 1199	*p*
Age Mean (SD), years < 65 65-74 ≥ 75	72.0 (11.5) 688 (23.2) 977 (32.9) 1306 (43.9)	74.9 (11.5) 162 (17.0) 270 (28.3) 521 (54.7)	70.3 (11.1) 217 (26.5) 296 (36.1) 306 (37.4)	70.8 (11.3) 309 (25.8) 411 (34.2) 479 (40)	< 0.0001 < 0.0001
Female, *n* (%)	1277 (43.0)	449 (47.1)	306 (37.4)	522 (43.5)	0.0002
**Type of atrial fibrillation**					
Paroxysmal	1488 (50.1)	460 (48.3)	411(50.2)	617 (51.5)	0.3384
Persistent	772 (26.0)	212 (22.2)	225 (27.5)	335 (27.9)	0.0059
Permanent	711 (23.9)	281 (29.5)	183 (22.3)	247 (20.6)	< 0.0001
Non-permanent	2260 (76.1)	672 (70.5)	636 (77.7)	952 (79.4)	< 0.0001
**Medical history**					
Hypertension	2512 (84.6)	805 (84.5)	709 (86.6)	998 (83.2)	0.1259
Heart failure	1892 (63.7)	653 (68.5)	494 (60.3)	745 (62.1)	0.0006
Vascular disease	1640 (55.2)	556 (58.3)	457 (55.8)	627 (52.3)	0.0181
Coronary artery disease	1467 (49.4)	495 (51.9)	410 (50.1)	562 (46.9)	0.0587
Previous myocardial infarction	641 (21.6)	244 (25.6)	163 (19.9)	234 (19.5)	0.0012
Peripheral artery disease	414 (13.9)	152 (15.9)	106 (12.9)	156 (13)	0.0929
Previous stroke/transient ischemic attack/peripheral embolism	488 (16.4)	146 (15.3)	148 (18.1)	194 (16.2)	0.2842
Diabetes mellitus	999 (33.6)	344 (36.1)	269 (32.8)	386 (32.2)	0.1400
Any previous bleeding	147 (4.9)	79 (8.3)	35 (4.3)	33 (2.8)	< 0.0001
Previous gastric bleeding	94 (3.2)	54 (5.7)	18 (2.2)	22 (1.8)	< 0.0001
Previous intracranial bleeding	16 (0.5)	8 (0.8)	4 (0.5)	4 (0.3)	0.2675
Malignancy	135 (4.5)	55 (5.8)	31 (3.8)	49 (4.1)	0.0831
**Thromboembolic risk**					
CHA_2_DS_2_-VASc score Mean (SD) =0 =1 ≥2	4.3 (1.8) 55 (1.9) 145 (4.9) 2771(93.3)	4.6 (1.7) 5 (0.5) 38 (4.0) 910 (95.5)	4.2 (1.9) 18 (2.2) 48 (5.9) 753 (91.9)	4.2 (1.9) 32 (2.7) 59 (4.9) 1108 (92.4)	< 0.0001 0.0013
**Bleeding risk**					
HAS-BLED score Mean (SD) ≥3	2.0 (0.9) 705 (23.7)	2.2 (0.9) 296 (31.1)	1.9 (0.9) 169 (20.6)	1.9 (0.8) 240 (20.0)	< 0.0001 < 0.0001
**Reason for hospitalization**					
Electrical cardioversion	764 (25.7)	151(15.8)	237 (28.9)	376 (31.3)	< 0.0001
Planned coronarography/PCI	282 (9.5)	83 (8.7)	79 (9.6)	120 (10.0)	0.5844
Planned CIED implantation/reimplantation	265 (8.9)	104 (10.9)	62 (7.6)	99 (8.3)	0.0281
Acute coronary syndrome	167 (5.6)	73 (7.7)	38 (4.6)	56 (4.7)	0.0041
Heart failure	595 (20.0)	279 (29.3)	138 (16.8)	178 (14.8)	< 0.0001
Ablation other than AF	144 (4.8)	28 (2.9)	43 (5.3)	73 (6.1)	0.0027
AF without any procedures	211 (7.1)	78 (8.2)	55 (6.7)	78 (6.5)	0.2829
Other	543 (18.3)	157 (16.5)	167 (20.4)	219 (18.3)	0.1042

The numbers are presented as the mean ± standard deviation, or as numbers (percentage) if otherwise mentioned. Abbreviation: AF, atrial fibrillation; CIED, cardiac implantable electronic device; IQR, interquartile range; SD, standard deviation. CHA_2_DS_2_-VASc score: congestive heart failure (1 point), hypertension (1 point), age ≥ 75 years (2 points), diabetes mellitus (1 point), stroke/TIA/thromboembolism (2 points), vascular disease (1 point), age 65–74 years (1 point), sex female (1 point). HAS-BLED score: hypertension (1 point), liver disease (1 point), renal disease (1 point), stroke history (1 point), bleeding history (1 point), age >65 years (1 point), and drug (concomitant use of NSAID or antiplatelet agent, 1 point).

**Table 2 jcm-09-03565-t002:** Results of the laboratory tests and echocardiographic examinations of patients treated with apixaban, dabigatran, and rivaroxaban. The numbers are presented as the mean ± standard deviation, or numbers (percentage) otherwise mentioned. Abbreviations: eGFR, estimated glomerular filtration rate; SD, standard deviation.

Clinical Characteristic	all *n* = 2971	Apixaban *n* = 953	Dabigatran *n* = 819	Rivaroxaban *n* = 1199	*p*
Laboratory tests					
Hemoglobin Mean (SD), g/dL	13.3 (1.8) *n* = 2942	12.8 (1.9) *n* = 942	13.5 (1.8) *n* = 809	13.5 (1.7) *n* = 1191	< 0.0001
White blood cell Mean (SD), K/μL	7.9 (3.1) *n* = 2935	8.0 (2.9) *n* = 939	7.9 (3.0) *n* = 807	7.9 (3.3) *n* = 1189	0.7752
Platelet Mean (SD), K/μL	220.1 (72.7) *n* = 2938	214.4 (69.8) *n* = 940	219.5 (66.5) *n* = 807	225 (78.5) *n* = 1191	0.0032
eGFR Mean (SD), ml/min/1.73m^2^	60.3 (20.2) *n* = 2947	54.1 (19.9) *n* = 945	64.1 (19.3) *n* = 811	62.6 (20.0) *n* = 1191	<0.0001
eGFR < 60 mL/min/1.73m^2^	1483 (50.3) *n* = 2947	581 (61.5) *n* = 945	354 (43.6) *n* = 811	548 (46.0) *n* = 1191	<0.0001
**Echocardiographic findings**					
Ejection fraction Mean (SD), %	49.5 (13.3) *n* = 2343	48.0 (14.1) *n* = 755	49.4 (13.0) *n* = 619	50.7 (12.6) *n* = 969	0.0002
Left atrial diameter Mean (SD), mm	46.5 (6.8) *n* = 2046	46.6 (7.0) *n* = 671	47.0 (6.8) *n* = 534	46.2 (6.6) *n* = 841	0.0796
Left ventricular systolic diameter Mean (SD), mm	39.4 (8.6) *n* = 1130	39.0 (9.6) *n* = 373	39.5 (8.6) *n* = 334	39.8 (7.6) *n* = 423	0.4639
Left ventricular diastolic diameter Mean (SD), mm	52.5 (8.2) *n* = 2230	52.3 (8.9) *n* = 737	52.8 (7.9) *n* = 578	52.5 (7.8) *n* = 915	0.4866

**Table 3 jcm-09-03565-t003:** Detailed data on anticoagulant therapy in the study group. Abbreviations: NOAC, non-vitamin K antagonist oral anticoagulant; VKA, vitamin K antagonist.

	All *n* = 2971	Apixaban *n* = 953	Dabigatran *n* = 819	Rivaroxaban *n* = 1199	*p*
Reduced dose	1071 (36.0)	334 (35)	325 (39.7)	412 (34.4)	0.0372
Antiplatelets with NOAC	399 (13.4)	141 (14.8)	96 (11.7)	162 (13.5)	0.1661
Treatment before hospitalization					
The same NOAC	2429 (81.8)	649 (68.1)	724 (88.4)	1056 (88.1)	< 0.0001
Another NOAC	82 (2.8)	62 (6.5)	13 (1.6)	7 (0.6)	< 0.0001
VKA	81 (2.7)	48 (5.0)	14 (1.7)	19 (1.6)	< 0.0001
Antiplatelets only	98 (3.3)	51 (5.4)	19 (2.3)	28 (2.3)	< 0.0001
None	281 (9.5)	143 (15.0)	49 (6.0)	89 (7.4)	< 0.0001

**Table 4 jcm-09-03565-t004:** Factors associated with the selection of an individual NOAC over another NOAC for stroke prevention in patients with AF, assessed using multivariable logistic regression models (participating centers were included as potential confounders).

Factors	Apixaban			Dabigatran			Rivaroxaban		
	OR	95%CI	*p*	OR	95%CI	p	OR	95%CI	*p*
Age (per 5 years)	1.14	1.09–1.19	<0.0001	0.95	0.91–0.99	0.0112	0.94	0.90–0.97	0.0009
Heart failure	1.38	1.14–1.67	0.001	0.84	0.70–1.03	0.0916	0.86	0.72–1.03	0.0958
Vascular disease	1.03	0.86–1.24	0.7241	1.20	0.99–1.45	0.0523	0.84	0.70–0.99	0.0464
Female	1.09	0.92–1.29	0.3223	0.8	0.67–0.96	0.0139	1.11	0.94–1.31	0.2033
Non-permanent AF	0.91	0.74–1.10	0.3319	1.03	0.83–1.27	0.812	1.09	0.90–1.33	0.3706
Malignancy	1.38	0.95–2.00	0.09	0.83	0.54–1.26	0.3737	0.86	0.59–1.26	0.4471
Any previous bleeding	2.37	1.67–3.38	<0.0001	0.86	0.57–1.28	0.4589	0.43	0.28–0.64	<0.0001
eGFR < 60 mL/min/1.73m^2^	1.38	1.15–1.64	0.0004	0.79	0.66–0.95	0.0108	0.91	0.77–1.08	0.2871
Antiplatelets with NOAC	1.19	0.91–1.55	0.1904	0.76	0.57–1.01	0.0564	1.08	0.84–1.39	0.5473

Abbreviation: AF, atrial fibrillation; CI, confidence interval; eGFR, estimated glomerular filtration rate; NOAC, non-vitamin K antagonist oral anticoagulant; OR, odds ratio.

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
