# Peer review of "Trends in the Prescription of Non-Vitamin K Antagonist Oral Anticoagulants for Atrial Fibrillation: Results of the Polish Atrial Fibrillation (POL-AF) Registry"

_jcm, 2020, doi:10.3390/jcm9113565_

Round 1

Reviewer 1 Report

I went through the paper by Iwona Gorczyca et al. entitled "Trends in the Prescription of Non-Vitamin K Antagonist Oral Anticoagulants for Atrial Fibrillation - Results of the POLish Atrial Fibrillation (POL-AF) Registry" several times. The article is pretty interesting and provides a good insight in NOAC's prescription in Poland. The overall length is quite good.

However, before being considered for publication, major revisions are needed:

1) what about Edoxaban prescription in Poland? Throughout the whole article there is no reference to this NOAC, please provide an explanation and add it in the discussion;

2) the important concept of the 5-fold increased ischemic stroke risk in AF patients, and the improved life expectancy and life quality in NOAC users should be reported (DOI: 10.3390/medicina55100617);

3) minor English spell check are required.

Author Response

Dear Reviewer,

I am pleased to resubmit for publication the revised version of Trends in the Prescription of Non-Vitamin K Antagonist Oral Anticoagulants for Atrial Fibrillation - Results of the POLish Atrial Fibrillation (POL-AF) Registry.

The reviewer’s comments were very helpful and greatly appreciated. We have addressed each concern and hope that this revised manuscript is now acceptable. Each comment is discussed in detail below. Revisions are indicated using a red font. Thank you for allowing us to resubmit our manuscript.

The specific responses to the editor’s comments are as follows:

1) what about Edoxaban prescription in Poland? Throughout the whole article there is no reference to this NOAC, please provide an explanation and add it in the discussion

Edoxaban has been registered in Europe as a drug preventing thromboembolic complications in patients with AF. However, it is not available in Poland. This explanation has been added in the Discussion section (line 221-222)

2) the important concept of the 5-fold increased ischemic stroke risk in AF patients, and the improved life expectancy and life quality in NOAC users should be reported (DOI: 10.3390/medicina55100617) (line 214-215)

The designated manuscript has been quoted and referred to in the Discussion section.

3) minor English spell check are required.

The article has been revised in terms of the language.

Thank you for the review and guidelines and we hope that now you will find our revised manuscript suitable for publication.

Kind Regards

Authors

Reviewer 2 Report

In the presented study "Trends in the Prescription of Non-Vitamin K Antagonist Oral Anticoagulants for Atrial Fibrillation - Results of the POLish Atrial Fibrillation (POL-AF) Registry", Gorczyca and co-workers investigate prescription patterns of oral anticoagulants in multiple centers in Poland. Of close to 4000 patients, 2971 patients were included in further analysis.

While Rivaroxaban was most commonly prescribed in about 40% of the patients, independent predictors were identified for treatment with apixaban (previous bleeding, GFR < 60 ml/min, old patients), rivaroxaban (non-permanent AF) and dabigatran (vascular disease).

Although the general impression of the manuscript is good, as it is well-written with few language errors and high scientific soundness, some issues have to be addressed to further improve the quality of the paper.

  1. Edoxaban beeing another NOAC, was not adressed in the present study. As Edoxaban is widely spread around the world, it should be either included in the study or adressed why it is not mentioned.
  2. In line 94, "admitted to hospital to have to AF ablation", there is a formulation error
  3. All patients undergoing AF ablation were excluded. Despite this issue is adressed and justified in the manuscript, exclusion of those patients seems to cut away a selected, but very important patient group in cardiology centers. This is a strong limitation of the study and should be mentioned in the limitations section.
  4. As about one third of patients in every drug group was treated with reduced dose, it should be investigated, if the reduction was according to the drug prescription guideline.
  5. Despite only 35% of patients with apixaban were treated with the reduced dose, patients in the apixaban group were significantly older and had a decreased renal function. As age, renal function/creatinine and body weight are determinants to choose apixaban dosage, body weight and body weight differences between the groups should be analyzed if applicable. Furthermore, it would be interesting, if patients were given apixaban only because they are only just in the reduced dose group being over 80 years of age or with low body weight or high age, as this is a potential confounder of apixaban treatment choice. Of note significantly more apixaban patients (6.5%) had another NOAC before hospitalisation, supporting this hypothesis.
  6. In the head line of Table 2, there is a spelling mistake "labaratory"

Author Response

Dear Reviewer,

I am pleased to resubmit for publication the revised version of Trends in the Prescription of Non-Vitamin K Antagonist Oral Anticoagulants for Atrial Fibrillation - Results of the POLish Atrial Fibrillation (POL-AF) Registry.

The reviewer’s comments were very helpful and greatly appreciated. We have addressed each concern and hope that this revised manuscript is now acceptable. Each comment is discussed in detail below. Revisions are indicated using a red font. Thank you for allowing us to resubmit our manuscript.

The specific responses to the editor’s comments are as follows:

  1. Edoxaban beeing another NOAC, was not adressed in the present study. As Edoxaban is widely spread around the world, it should be either included in the study or adressed why it is not mentioned.

Edoxaban has been registered in Europe as a drug preventing thromboembolic complications in patients with AF. However, it is not available in Poland. This explanation has been added in the Discussion section (line 221-222).

  1. In line 94, "admitted to hospital to have to AF ablation", there is a formulation error

The indicated fragment has been corrected to "admitted to hospital to have AF ablation"

  1. All patients undergoing AF ablation were excluded. Despite this issue is adressed and justified in the manuscript, exclusion of those patients seems to cut away a selected, but very important patient group in cardiology centers. This is a strong limitation of the study and should be mentioned in the limitations section.

Patients admitted to hospital to have AF ablation were excluded from the registry because of two reasons. Firstly, not all the centers perform catheter ablation. Secondly, it was acknowledged that patients undergoing ablation due to AF have a clinical profile different from most patients with AF (they are younger and do not have concomitant diseases). It has been described in Limitations section (line 302-305).

  1. As about one third of patients in every drug group was treated with reduced dose, it should be investigated, if the reduction was according to the drug prescription guideline.

Among 1071 patients who were treated with the reduced dose of NOACs, inappropriately reduced doses were observed in 242 patients (22.6%). The above data has been included in the Results section (line 178-180).

  1. Despite only 35% of patients with apixaban were treated with the reduced dose, patients in the apixaban group were significantly older and had a decreased renal function. As age, renal function/creatinine and body weight are determinants to choose apixaban dosage, body weight and body weight differences between the groups should be analyzed if applicable. Furthermore, it would be interesting, if patients were given apixaban only because they are only just in the reduced dose group being over 80 years of age or with low body weight or high age, as this is a potential confounder of apixaban treatment choice. Of note significantly more apixaban patients (6.5%) had another NOAC before hospitalisation, supporting this hypothesis.

In the study group, 36% of patients were treated with a reduced NOAC dose. The reduced dose for stroke prevention was used in 35% of apixaban patients, 39.7% of dabigatran patients, and 34.4% of rivaroxaban patients (p=0.037). In the study group appropriate NOAC dose reduction was observed in 761 patients (71.1%) and inappropriate NOAC dose reduction was observed in 242 patients (22.6%), in 68 patients (6.3%) there were no data allowing to assess the appropriateness of the reduced NOAC dose choice.

An inappropriately reduced dose was observed in 120 patients (29.1%) treated with rivaroxaban, in 93 patients (27.8%) treated with apixaban and in 29 patients (8.9%) treated with dabigatran (p<0.0001).

In another manuscript, which is currently being revised, the POL-AF registry researchers evaluated the frequency of inappropriately reduced NOAC dose prescription and also the factors predisposing to such choices.

  1. In the head line of Table 2, there is a spelling mistake "labaratory"

The indicated mistake has been corrected (line 164).

Thank you for the review and guidelines and we hope that now you will find our revised manuscript suitable for publication.

Kind Regards

Authors

Reviewer 3 Report

This study evaluates NOAC prescription in a large cohort of patients comparing baseline parameters between individual prescription groups. The manuscript by Gorczyca et al. mentions the important question as to which NOAC is most suitable for which patient subgroup. However, the actual study describes overall manners of prescription rather than evaluating clinical outcomes associated with the respective therapies. Prescription habits in hospitalized patients may be influenced by subjective physicians's preferences or standards of the respective centres. Therefore, it should be tested in how far the contributing centre may act as a confounder.

In particular the Discussion section of manuscript requires extensive editing of English language and scientific style.

Overall, the manuscript is of low clinical interest as clinically relevant endpoints are not evaluated.

Author Response

Dear Reviewer,

I am pleased to resubmit for publication the revised version of Trends in the Prescription of Non-Vitamin K Antagonist Oral Anticoagulants for Atrial Fibrillation - Results of the POLish Atrial Fibrillation (POL-AF) Registry.

The reviewer’s comments were very helpful and greatly appreciated. We have addressed each concern and hope that this revised manuscript is now acceptable. Each comment is discussed in detail below. Revisions are indicated using a red font. Thank you for allowing us to resubmit our manuscript.

The specific responses to the editor’s comments are as follows:

1. However, the actual study describes overall manners of prescription rather than evaluating clinical outcomes associated with the respective therapies. Prescription habits in hospitalized patients may be influenced by subjective physicians's preferences or standards of the respective centres. Therefore, it should be tested in how far the contributing centre may act as a confounder.

According to the currently ESC guidelines from 2020 and AHA guidelines from 2019 concerning treatment of patients with AF, individual NOACs are not recommended for particular groups of AF patients. Lack of NOAC positioning for patients with AF can on the one hand make it difficult to choose a particular drug, and on the other hand offers a wide variety of choice. Due to that the POL-AF registry researchers decided that it would be interesting to analyse NOAC choice predictors. Additionally to the above mentioned guidelines, there are expert documents available which suggest the choice of  individual NOACs in particular clinical situations. However, they do not have a guideline status and they are not consistent. The presented article can be considered as an attempt to show whether clinicians take into account expert documents when they make their decisions.

2. In particular the Discussion section of manuscript requires extensive editing of English language and scientific style.

The Discussion section has been edited.

Thank you for the review and guidelines and we hope that now you will find our revised manuscript suitable for publication.

Kind Regards

Authors

Round 2

Reviewer 1 Report

The authors improved the article according to my suggestions, it can be accepted in present form for publication.

Author Response

Dear Reviewer,

Thank you for acceptance for publication the revised version of Trends in the Prescription of Non-Vitamin K Antagonist Oral Anticoagulants for Atrial Fibrillation - Results of the POLish Atrial Fibrillation (POL-AF) Registry.

Kind Regards

Authors

Reviewer 3 Report

The authors have attempted to improve the overall manuscript with regard to writing style. The focus on the analysis of prescription habits has been clarified. Still, clinical endpoints (stroke, bleeding complications, etc.) would increase the relevance of the data and the manuscript. Statistical analysis to correct for participating centers as potential confounders with respect to prescription habits should be attempted. Still, the manuscript should be revised by a native speaker and improved as to scientific writing style.

Author Response

Dear Reviewer,  

I am pleased to resubmit for publication the revised version of Trends in the Prescription of Non-Vitamin K Antagonist Oral Anticoagulants for Atrial Fibrillation - Results of the POLish Atrial Fibrillation (POL-AF) Registry.

The reviewer’s comments were very helpful and greatly appreciated. We have addressed each concern and hope that this revised manuscript is now acceptable. Each comment is discussed in detail below. Revisions are indicated using a red font. Thank you for allowing us to resubmit our manuscript.

The specific responses to the editor’s comments are as follows:

- Still, clinical endpoints (stroke, bleeding complications, etc.) would increase the relevance of the data and the manuscript.

The aim of this study was to assess the frequency of using apixaban, dabigatran, and rivaroxaban, and the predictors of their prescription in a nationwide cohort of hospitalized patients with AF. According to the aims of the study no follow-up was planned in the researched group (NCT04419012). Assessment of the impact of applied anticoagulation therapy on long-term prognosis is the aim of a further study which will be made by the group of POL-AF registry researchers.

- Statistical analysis to correct for participating centers as potential confounders with respect to prescription habits should be attempted.

Statistical analysis has been corrected. Participating centers have been included in multivariable logistic regression models as potential confounder. Table S1 has been added. It shows the proportion of patients treated with apixaban, dabigatran and rivaroxaban in particular centers. Local conditions influencing application of a particular NOAC have been described in the Discussion section.

- Still, the manuscript should be revised by a native speaker and improved as to scientific writing style.

The manuscript has been edited by a native speaker. I enclose a certificate confirming the language proofreading.

Thank you for the review and guidelines and we hope that now you will find our revised manuscript suitable for publication.

Kind Regards

Authors
